# PRIVACY-AWARE HYBRID IMAGE SYNTHESIS WITH LOCAL-CLOUD COLLABORATION

## ABSTRACT

Recent advances in diffusion-based generative models have enabled high-quality and personalized image synthesis. However, protecting user privacy while enabling efficient image generation remains a major challenge when deploying diffusion models on edge devices. Cloud-based inference risks exposing sensitive content, whereas fully local execution demands excessive computation and memory. This calls for a collaborative edge–cloud paradigm that can balance both concerns. In this work, we propose *PrivInfer*, a training-free, privacy-preserving inference framework that enables efficient edge-cloud collaboration for image generation. *PrivInfer* decomposes the generation process by region: privacy-sensitive areas are processed locally, while non-sensitive regions are offloaded to the cloud. This design reduces on-device computation while minimizing privacy risks. To ensure secure cross-device interaction, we introduce a more secured mechanism that shares structural information without exposing raw features. We further develop a ring-based masking strategy to structurally isolate private content during convolution, and a heterogeneous-step scheme that enables low-step local models to leverage high-fidelity cloud features. Extensive experiments show that *PrivInfer* significantly reduces inference steps and computational load on edge devices, while maintaining high generation fidelity and strong privacy protection, which offers a practical solution for private and efficient diffusion model deployment.

## 1 INTRODUCTION

The rapid development of generative models, particularly diffusion models (Esser et al., 2024; Labs, 2023; Gao et al., 2024; Song et al., 2020a), has significantly advanced the capabilities of image editing and synthesis. These models (Ruiz et al., 2023; Ye et al., 2023; Kumari et al., 2023; Nichol et al., 2021; Zhang et al., 2023) enable high-fidelity generation, flexible content manipulation, and structure-aware editing, empowering applications such as image inpainting, face swapping, style transfer, and personalized visual content creation. As generative models become increasingly integrated into cloud services and consumer devices, they offer unprecedented opportunities for personalized creative expression, content restoration, and intelligent vision applications (Ye et al., 2023; Gal et al., 2022; Zhang et al., 2023; Tan et al., 2024).

However, the same capabilities also introduce new challenges in privacy protection (Pittaluga & Zhuang, 2023; Shamshad et al., 2023). Many real-world visual inputs include privacy-sensitive regions, such as faces, exposed skin, or identifying details, mixed with non-sensitive contextual information, such as background, clothing, or objects. Uncontrolled image uploading, editing, or sharing can lead to privacy leakage, particularly when all data is processed by centralized, cloud-based generative models. In contrast, fully local processing ensures maximal privacy, but it is often hindered by the high computational and memory demands of modern diffusion models. These typically exceed the capabilities of edge devices such as smartphones or browsers, leading to slow execution, and degraded image quality.

While many existing works have explored model distillation (Lin et al., 2024; Salimans & Ho, 2022; Luo et al., 2023; Sauer et al., 2024) or pruning (Fang et al., 2023; Castells et al., 2024; Wang et al., 2024) to enable efficient image generation on resource-constrained devices, these approaches typically require task-specific fine-tuning, or access to much training data, which significantly increases the development cost. Moreover, the process of compressing or distilling large generative models

(a) Original generative method  (b) Efficient and privacy-secure image generation

Figure 1: (a) Original generative method. (b) Efficient and privacy-secure image generation proposed in our work. Our local model operates with minimal input conditions and fewer inference steps, significantly reducing computational cost.

often leads to performance degradation, especially in fine-grained editing tasks where high-fidelity outputs are crucial. While caching-based (Ma et al., 2024b;a; Kahatapitiya et al., 2024) methods leverage feature similarity across timesteps to reduce redundant computation, they typically rely on a consistent inference schedule and operate on the full image, limiting their flexibility. All these limitations motivate the development of **a training-free method that operates with both fewer inference steps and reduced computational cost on edge devices**.

Unlike many prior works that aim to shrink the entire model for deployment (Lin et al., 2024; Luo et al., 2023; Sauer et al., 2024; Fang et al., 2023; Wang et al., 2024), in this work, we argue that not all regions in an image require equal treatment: only private areas need strict on-device protection, while the rest can benefit from high-capacity, cloud-based generation, as shown in Fig. 1. This eliminates the need for retraining and allows us to leverage powerful pre-trained models with less computational cost. However, a naïve separation of the generation process may lead to inconsistencies in the generated content, while directly sharing feature representations (e.g., keys and values) poses a significant risk of leaking private attributes.

Our key observation is that it is sufficient to exchange Gram matrices rather than full key and value maps to compute global attention within linear attention mechanisms. By transmitting and accumulating Gram matrices alone, accurate global attention scores can be computed while eliminating the exposure of sensitive features. To complement this, we also introduce a ring-based masking strategy that structurally isolates private content in convolutional operations, achieving strong and explicit privacy guarantees. Furthermore, utilizing the similarity between adjacent denoising steps, we use cached features from previous steps and adopt heterogeneous-step asynchronous communication to reduce the communication latency. We implemented a **Priv**acy-aware asynchronous interaction framework that supports heterogeneous **Infer**ence steps across devices, called *PrivInfer*.

*PrivInfer* only requires off-the-shelf pre-trained models. With highly reduced steps as well as computational cost on edge devices, it can still achieve performance comparable to the original model. We summarize our main contributions as follows:

- We provide new insights into privacy-aware image editing. Region-wise separation allows the client to perform generation and editing on privacy-sensitive areas using fewer inference steps and reduced spatial scope, thereby enhancing efficiency while preserving privacy.

- A privacy-preserving asynchronous inference framework *PrivInfer* is proposed, which exchanges Gram matrices instead of full key-value features. We also propose a ring-based masking strategy that isolates private content during convolutional operations.

- Extensive experiments are conducted and show that *PrivInfer* significantly reduces local inference steps and computational cost on edge devices, while preserving both generation quality and strong privacy guarantees through efficient cross-device collaboration. Both qualitative and quantitative experimental results validate the effectiveness of our method.

## 2 RELATED WORK

### 2.1 IMAGE SYNTHESIS

Image synthesis has evolved through several major classes of generative models: GANs, VAEs, autoregressive models, and diffusion models. GANs (Karras, 2019; Brock, 2018; Goodfellow et al.,

2014) are known for producing high-quality images but suffer from training instability and mode collapse. VAEs (Kingma, 2013; Sohn et al., 2015; Higgins et al., 2017; Reed et al., 2016) provide a structured latent space and stable optimization, though often at the cost of image sharpness. Autoregressive models (Van Den Oord et al., 2016; Sun et al., 2024; Tian et al., 2024; Yu et al., 2024) generate images pixel-by-pixel or token-by-token with high fidelity but are computationally expensive. Recently, diffusion models (Esser et al., 2024; Labs, 2023; Gao et al., 2024; Ho et al., 2020; Sohl-Dickstein et al., 2015; Song et al., 2020b) still serve as the leading paradigm, offering strong performance and high quality image generation. However, their iterative sampling process is slow, prompting a surge of research into acceleration techniques.

## 2.2 DIFFUSION ACCELERATION

Although diffusion models have demonstrated impressive performance in image synthesis, their inference typically requires plenty of iterative steps, resulting in high computational cost and latency. To address this, numerous methods have been proposed to accelerate diffusion-based generation. Denoising acceleration methods, such as DDIM (Song et al., 2020a) and DPM-Solver (Lu et al., 2022), reduce the number of required steps by improving the integration schemes for sampling. Other approaches leverage knowledge distillation, such as progressive distillation (Salimans & Ho, 2022) and consistency distillation (Song et al., 2023). More recent work explores caching and reusing intermediate features to avoid redundant computation across steps, including work like DeepCache (Ma et al., 2024b). Despite these efforts, many acceleration methods require either retraining or sacrificing image quality. Our work builds upon this line of research, targeting both step reduction and privacy-awareness without requiring model fine-tuning.

## 2.3 PRIVACY PROTECTION

With the increasing deployment of image generation and editing models on edge devices and personal data, privacy preservation has become a critical concern. Traditional privacy protection methods include image obfuscation techniques such as blurring and pixelation McPherson et al. (2016), offers quick privacy protection but significantly reduces image quality and utility. In contrast, learning-based anonymization using generative networks (Sun et al., 2018; Hukkelås et al., 2019), enables realistic yet anonymized images, preserving key attributes while obscuring identifiable features. Adversarial perturbation techniques (Li et al., 2023; Hsieh & Li, 2021; Low et al., 2022) deceive deep learning models through small, carefully designed modifications, and have been applied in privacy protection.

## 3 METHODS

In this section, we describe our proposed method *PrivInfer* in detail. Firstly, some preliminaries related to flow-based diffusion models are introduced in Section 3.1. Then in Section 3.2, we delve deeper into attention interactions, propose to protect privacy features by propagating Gram matrices. Section 3.3 proposes a ring-based masking strategy, and in Section 3.4, we introduce a heterogeneous-step interaction method, in which high-quality results generated on the server side can be used to guide the inference of local models.

## 3.1 PRELIMINARY

Diffusion models generate images by learning to reverse a stochastic process that gradually transforms data into noise. This process defines a continuous trajectory from structured data to pure noise, and the generative model learns to invert this trajectory.

Formally, the forward (noising) process is described by a stochastic differential equation (SDE):

$$dx_t = f(x_t, t)dt + \sigma(t)dW_t \tag{1}$$

where $f(x_t, t)$ represents the deterministic evolution of the data (drift), $\sigma(t)$ controls the time-dependent noise magnitude, and $W_t$ is the standard Wiener process introducing randomness.

A common training objective is to minimize the error between the injected noise and the predicted noise at randomly sampled time steps. Alternative perspectives, such as flow-based approaches (Lipman et al., 2022; Liu et al., 2022; Xie et al., 2024), treat generation as evolving probability densities

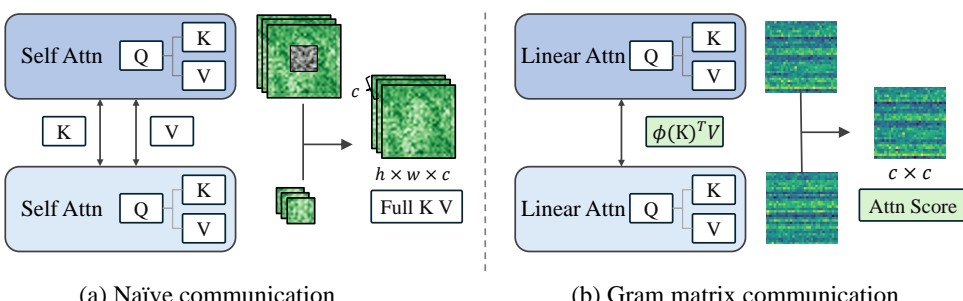

(a) Naïve communication              (b) Gram matrix communication

Figure 2: (a) Naïve communication between the server and the local. (b) Our proposed gram matrix communication. Our method transmits less information and avoids exposing the K and V features.

along a vector field, with training based on matching the learned velocity field $v_t(x)$ to the true dynamics:

$$\mathcal{L}_{flow} := \mathbb{E}_{x,t}\left[\|v_t(x) - u_t(x)\|_2^2\right] \tag{2}$$

By reformulating the learning dynamics, this flow-based approach explicitly encourages a straightened mapping between noise and image space, thereby enabling faster generation of high-quality images during inference.

While better trajectory can be achieved through training strategies, reducing the number of inference steps via model interaction mechanisms at inference time still represents a valuable and underexplored research direction with strong practical implications, particularly for realistic on-device applications.

## 3.2 GRAM MATRIX COMMUNICATION

The key objective of *PrivInfer* is to realize efficient and privacy-secure cooperation between the server and local client for image synthesis. Naïvely, this can be achieved by (1) separating private and non-private regions for computation on the client and server respectively; or (2) synchronizing intermediate activations at aligned timesteps. While handling the two regions separately alleviates local computation, such separation can introduce semantic inconsistencies, which calls for necessary cross-region communication. As for (2), previous methods (Li et al., 2024b; Shih et al., 2023) have explored parallelize computation across devices to facilitate high-resolution generation. However, model interaction inherently involves transmitting full intermediate features between devices, which in practice, as for self-attention layers, refers to the exchange of Keys (K) and Values (V). In essence, K and V reflect detailed layer-wise representations of the image. Consequently, if the model parameters are accessible, this results in full leakage of private content, which runs counter to the privacy-preserving motivation behind region-wise separation, as shown in Fig. 2 (a).

To address this issue, we propose leveraging attention score interaction to avoid the leakage of private information, as shown in Fig. 2 (b). Fortunately, the design of linear attention in diffusion models provides a favorable foundation for this approach. The standard attention mechanism is typically implemented as follows:

$$\text{Attention}(\mathbf{Q}, \mathbf{K}, \mathbf{V}) = \text{Softmax}(\frac{(\mathbf{Q}\mathbf{K}^\top)}{\sqrt{d}})\mathbf{V}, \tag{3}$$

where $\mathbf{Q} \in \mathbb{R}^{l \times c}$, $\mathbf{K} \in \mathbb{R}^{l' \times c}$, $\mathbf{V} \in \mathbb{R}^{l' \times c'}$ are the query, key, and values of the attention operation respectively. Following standard practice, we adopt $l' = l$, $c' = c$ throughout the paper. In this setting, due to the presence of the *Softmax* operation, it necessitates full access to K and V in order to compute attention scores and produce the final attended representations. In addition, it incurs a computational cost that scales quadratically with the input length, posing a significant bottleneck when handling long sequences. To address the above issue, many models (Liu et al., 2024; Gu & Dao, 2023; Xie et al., 2024; Yang et al., 2023) reformulate the attention computation to achieve linear complexity with respect to input length. Intrinsically, they approximate the softmax operation as a kernel decomposition, which facilitates a reformulation of attention:

$$\text{Softmax}(\mathbf{Q}, \mathbf{K}) \approx \phi(\mathbf{Q})\,\phi(\mathbf{K})^T \tag{4}$$

such that

$$\text{Attention}(\mathbf{Q}, \mathbf{K}, \mathbf{V}) \approx \phi(\mathbf{Q}) \, (\phi(\mathbf{K})^T \, \mathbf{V}) \tag{5}$$

Therefore, we may first calculate an intermediate component $\phi(\mathbf{K})^T \, \mathbf{V}$ as the attention score, and then apply it to the projected $\mathbf{Q}$. Under this setting, the resulting attention score is of shape $c \times c$, representing the channel-wise correlations as a Gram matrix.

Notably, we observe that attention scores computed via linear attention are point-wise additive. Therefore, our design focuses on transmitting only the attention scores, which are then added point by point to obtain the full attention score, as presented in the Fig. 2 (b). After getting partial attention score, the full $\phi(\mathbf{K})^T \, \mathbf{V}$ is obtained through:

$$\phi(\mathbf{K})^T \, \mathbf{V} = (\phi(\mathbf{K})^T \, \mathbf{V})_{non-privacy} + (\phi(\mathbf{K})^T \, \mathbf{V})_{privacy} \tag{6}$$

Unlike the naïve approach in Figure (a), ours avoids explicit transmission of partial K and V, thus prevents the direct leakage of features from the private regions.

Moreover, after the cloud model acquires the transmitted Gram matrix of the features, due to the inherent ambiguity in the Gram matrix, it is infeasible for it to retrieve the explicit private K and V information. Specifically, for gram matrix S, if X is a solution that satisfies $\mathbf{X}^T\mathbf{X} = \mathbf{S}$, then for any orthogonal matrix $\mathbf{Q}$, it always holds:

$$\mathbf{S} = \mathbf{X}^T\mathbf{X} = (\mathbf{X}\mathbf{Q})^T\mathbf{X}\mathbf{Q} \tag{7}$$

This implies that X can at most be reconstructed up to an orthogonal transformation, corresponding to its equivalence class:

$$[\mathbf{X}] = \{\mathbf{X}\mathbf{Q} \mid \mathbf{Q} \in \mathbb{R}^{c \times c}, \mathbf{Q}^T\mathbf{Q} = \mathbf{I}\} \tag{8}$$

As a result, only the projection relationships between vectors are accessible, and the exact feature vectors cannot be recovered, thus ensuring user privacy to the greatest extent.

## 3.3 RING-BASED MASKING STRATEGY

Beyond attention mechanisms, for each layer, it always involves MLP operations. In practice, this is frequently realized using convolution, aiming to improve the spatial consistency among image regions, like in Podell et al. (2023); Xie et al. (2024). To prevent potential privacy leakage during convolution operations, we also introduce a ring-based masking strategy that structurally isolates private content in convolutional operations.

As shown in the input of Fig. 3, for each input condition, we have a clearly defined private region and we extend it with a ring-shaped area, denoted in pink, to accommodate convolutional processing. This expanded region is fed into the local model, while the external non-private portion is handled by the server. If the convolution kernel is of size 3 or less, the expansion involves one patch; with a kernel size of 1, no expansion is required.

Owing to the local nature of convolution, the server only needs to transmit the blue ring area outside the pink region back to the client. Similarly, feature transmission from the client is limited to the pink-ring area. Consequently, during each convolutional interaction, only features from the pink and blue rings are exchanged. And since the pink ring lies beyond the core privacy region, no private content is exposed during communication.

## 3.4 HETEROGENEOUS-STEP ASYNCHRONOUS INTERACTION

Although synchronization at aligned timesteps can mitigate inconsistency across generated regions, it introduces substantial latency overhead. To address this, we exploit the high similarity of features across adjacent steps by enabling interaction between the current state and cached features from previous timesteps.

In addition, due to heterogeneous device capabilities, the server and client often operate at different speeds. Rather than enforcing strict aligned steps, we exploit this disparity by allowing the server to perform more inference steps. The resulting higher-quality representations from the server can subsequently be used to guide the local model, enabling it to achieve comparable performance with substantially fewer steps.

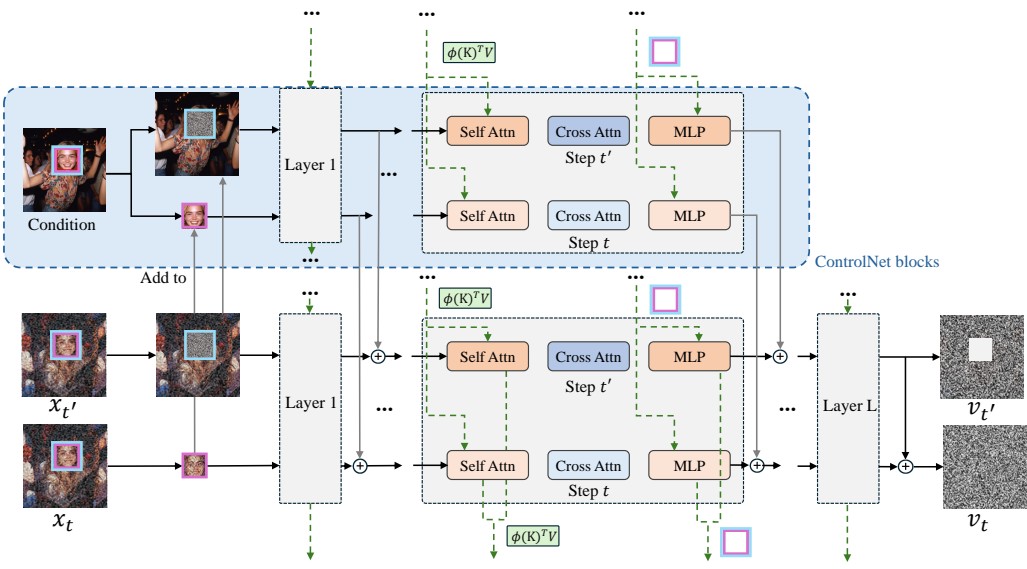

Figure 3: Overview of *PrivInfer*. It is a privacy-aware asynchronous interaction framework that supports heterogeneous Inference steps across devices.

We implemented a **Priv**acy-aware asynchronous interaction framework that supports heterogeneous **Infer**ence steps across devices, called *PrivInfer*. Fig. 3 illustrates the asynchronous interaction between the server and the local model in cooperation with ControlNet. The server and client are at different inference steps at the same timestamp. At each step, both sides utilize the cached features from the previous communication to compute the complete attention scores and convolution ring regions. The system then determines whether the current features need to be stored for future transmission. Only at time steps requiring asynchronous communication are newly generated features saved for transmission. Moreover, the transmission process is asynchronous with respect to computation, thus the communication overhead is effectively hidden within the computation pipeline.

## 4 EXPERIMENTS

### 4.1 SETUPS

**Models** Our method only requires off-the-shelf models, making it easy to integrate with existing frameworks. Since our framework leverages linearized attention, we conduct our experiments using the recently proposed SANA Xie et al. (2024) model. SANA is a text-to-image framework that can efficiently generate high resolution images because of the design of linear DiT and deep compression autoencoder with the scaling factor 32. And for image-to-image tasks, we conducted experiments on a ControlNet-augmented SANA model, conditioned on input HED maps. We use the 0.6B version of SANA unless mentioned.

**Datasets** We first conducted text-to-image experiments to demonstrate the feasibility of our approach, followed by image-to-image experiments to evaluate the effectiveness of conditional generation based on image inputs. For the text-to-image experiments, we used the MJHQ-30K (Li et al., 2024a) dataset for performance evaluation and visualization. And for the image-to-image setting, we use the COCO 2017 validation set, which has 5,000 images with bounding box (bbox) and image captions.

**Metrics** Following previous method (Meng et al., 2021; Li et al., 2022; Xie et al., 2024), we employ Fréchet Inception Distance (FID) (Heusel et al., 2017), Peak Signal-to-Noise Ratio (PSNR), and Learned Perceptual Image Patch Similarity (LPIPS) (Zhang et al., 2018) to assess generation quality. FID quantifies the divergence in distribution between generated samples and reference images, while PSNR evaluates pixel-wise similarity, particularly between our outputs and those of the original

Table 1: Quantitative results on the MJHQ-30K dataset. The generated images are of size 512×512. w/ G.T. means calculating the metrics with the ground-truth images, while w/ Orig. means with the original model's samples, i.e., with 20 inference steps. At the original steps, our method effectively preserves performance. Under reduced steps, it significantly outperforms single-device inference.

| #Steps | Method | PSNR (↑) | LPIPS (↓) | | FID (↓) | | ImageReward (↑) | CLIP-T (↑) |
| | | | w/ G.T. | w/ Orig. | w/ G.T. | w/ Orig. | | |
|---|---|---|---|---|---|---|---|---|
| 20 | Original | – | 0.678 | – | 6.85 | – | 1.11 | 28.28 |
| 20 | Naïvely Separation | 15.08 | 0.684 | 0.362 | 14.06 | 10.53 | 0.56 | 27.23 |
| | **Ours** | **21.66** | **0.678** | **0.131** | **6.84** | **0.92** | **1.09** | **28.17** |
| 5 | Single device | 14.96 | 0.695 | 0.411 | 14.97 | 9.73 | 0.76 | 27.50 |
| | **Ours** | **16.59** | **0.682** | **0.283** | **8.32** | **3.36** | **0.97** | **27.84** |

model. LPIPS serves as a perceptual similarity measure to reflect human visual perception. Besides, we also use ImageReward (Xu et al., 2023) to evaluate human preference and CLIP-T (Hessel et al., 2021) metric to evaluate prompt alignment.

**Implementation details**   We adopt the 20-step Flow-DPM-Solver with guidance scale 4.5 as the baseline inference setting on the server-side model throughout all experiments on SANA. In contrast, the client-side model is evaluated under two configurations with 20 and 5 inference steps. For the text-to-image experiments, results are generated at 512x512 on the MJHQ-30K dataset. We designate the central region of the image as the privacy-sensitive area to facilitate feasibility validation and qualitative assessment, enabling a clear partition between server-side and client-side processing. As for image-to-image generation, we extract HED maps from COCO images, resize them to 1024×1024, and apply the SANA 0.6B-based ControlNet model for conditional synthesis. The area within the bounding box of each image is defined as the privacy region.

## 4.2   MAIN RESULTS

**Text-to-image results**   Table 1 presents the quantitative results on the MJHQ-30K dataset. For this text-to-image task, the generated images are of size 512×512. We can see that with the full inference steps, our method maintains comparable performance to the baseline. When the number of steps is reduced, it substantially improves results compared to single-device inference. Fig. 5 also significantly validates the effectiveness of our method. We observe that the results generated by Naïve Separation exhibit clear boundary inconsistencies, indicating a failure to maintain coherence between private and non-private regions. In contrast, our method achieves strong alignment with the original image even under asynchronous communication with only 20 inference steps. Furthermore, when the number of local steps is reduced to just 5, traditional single-device inference yields only coarse, low-quality results. However, benefiting from high-quality server-side guidance, our method effectively denoises private regions and harmonizes boundary transitions, resulting in high-quality generation across the entire image.

**Image-to-image results**   Table 3 presents the quantitative results on the COCO 2017 validation dataset. For this image-to-image task, the generated images are of size 1024x1024. Compared to single-device inference, our approach achieves a higher degree of similarity to the results produced by full-step (original) inference. As shown in Fig. 6, while 5-step inference on a single device results in significant quality loss, our method overcomes this limitation by leveraging external guidance, achieving sharp, detailed, and high-quality image generation under the same low-step setting. In addition, as shown in Fig. 6 (c), the server-side outputs do not reveal any information within the protected regions.

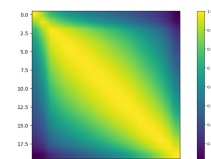

Figure 4: Similarity between steps.

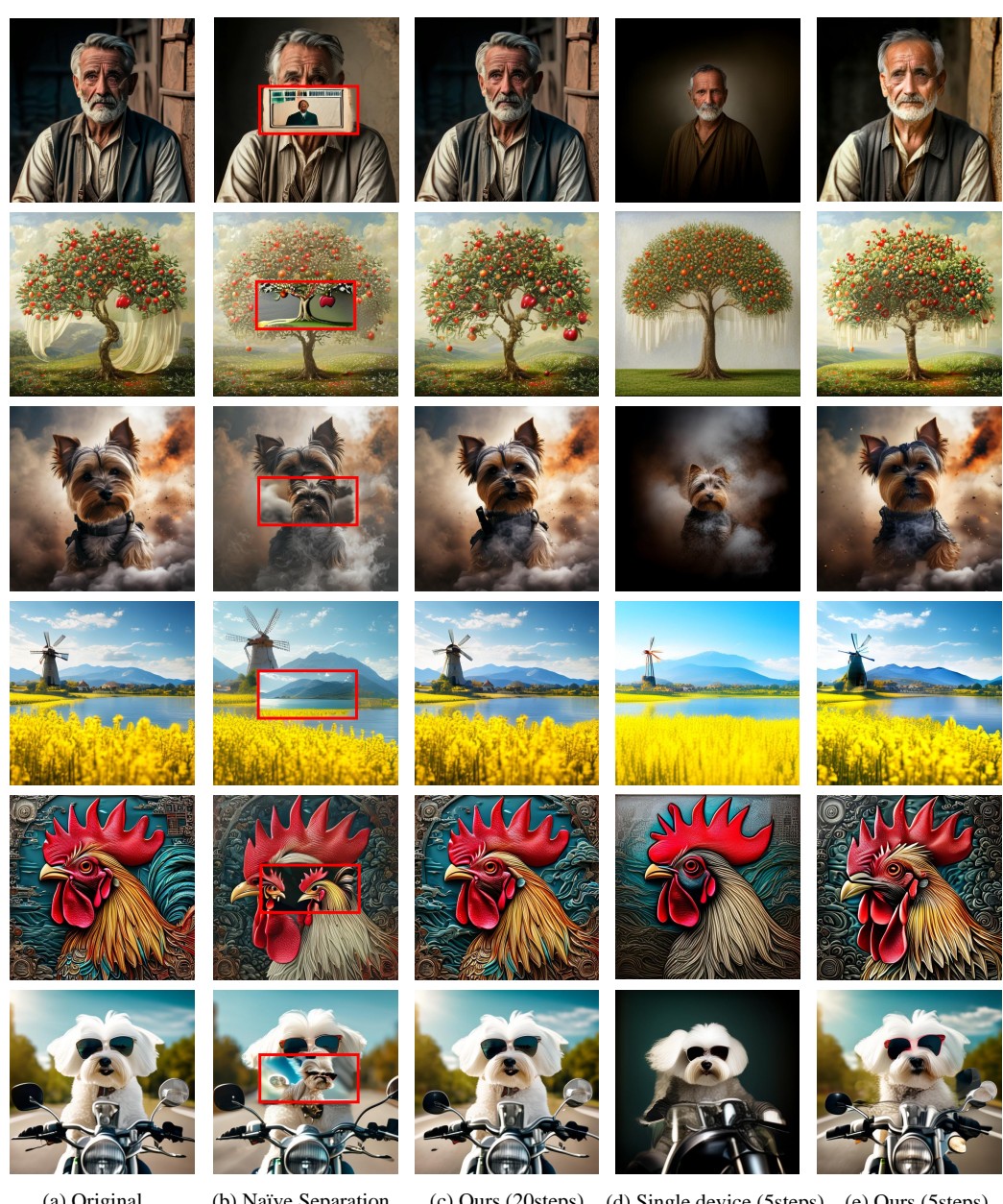

(a) Original     (b) Naïve Separation     (c) Ours (20steps)     (d) Single device (5steps)     (e) Ours (5steps)

Figure 5: Qualitative results on the MJHQ-30K dataset. The generated images are of size 512×512.

### 4.3 ABLATION STUDY

**Similarity between step features** We visualize the step-wise similarity across averaged layers, as shown in Fig. 4. Features between adjacent layers exhibit high similarity, which allows us to reuse cached features for asynchronous transmission. Moreover, compared to raw features, the linearized attention scores computed in our method aggregate statistical information across the channel dimension, resulting in greater stability and consistency across inference steps. Further details are presented in the appendix.

**Feature communication** As shown in Fig. 5 (b), if the cloud and local models perform inference independently without interaction, the generated content in the two regions remains disjoint, resulting in visu-

Table 2: Results of computational cost.

| Method | MACs (T) ↓ |
|---|---|
| Original | 21.65 |
| Ours | **0.84** |

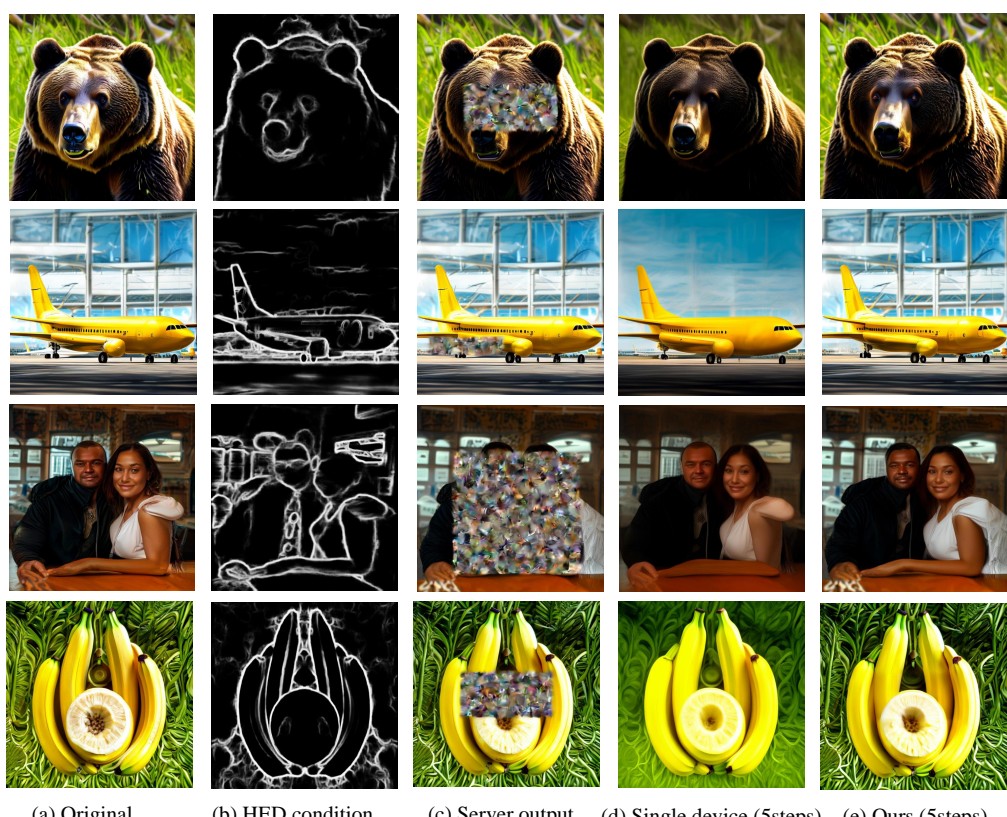

(a) Original     (b) HED condition     (c) Server output     (d) Single device (5steps)     (e) Ours (5steps)

Figure 6: Qualitative results on the COCO 2017 dataset. The generated images are of size 1024x1024.

Table 3: Quantitative results on the COCO 2017 dataset. Under reduced steps, our method yields results that are more similar to the original full-step inference compared to single-device inference.

| #Steps | Method | PSNR (↑) | LPIPS (↓) | |
|---|---|---|---|---|
| | | | w/ G.T. | w/ Orig. |
| 20 | Original | – | 0.706 | – |
| 5 | Single device | 15.80 | **0.685** | 0.316 |
| | **Ours** | **19.22** | 0.717 | **0.198** |

ally inconsistent and incoherent images. In contrast, our method mitigates this issue by introducing asynchronous communication that effectively synchronizes the generation process.

**Few-step in local devices** On the COCO 2017 validation set, the average area ratio of the utilized bounding boxes is estimated to be 11.11%. This enables an approximate estimation of the total MACs incurred during inference. As summarized in Table 2, due to the use of fewer steps and smaller inputs, our approach achieves a substantial reduction in local inference computation. In real-world scenarios, the privacy regions are often even smaller, which allows for further reduction in computational cost. This reveals new potential for local model deployment and inference.

## 5 CONCLUSION

We propose *PrivInfer*, a training-free, privacy-aware framework for efficient local-cloud collaborative image generation. By separating private and non-private regions, and exchanging only Gram matrices for attention, *PrivInfer* preserves sensitive content while reducing local computation. With ring-based masking and heterogeneous-step interaction, it achieves strong privacy guarantees and high-quality outputs without retraining. Experiments confirm its effectiveness and practicality for privacy-sensitive generative tasks.

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

## A MORE RESULTS

Fig. 7 shows more visualization results on the MJHQ-30K dataset. It can be observed that our model achieves a generation quality close to the original image using only 5 inference steps on the local device.

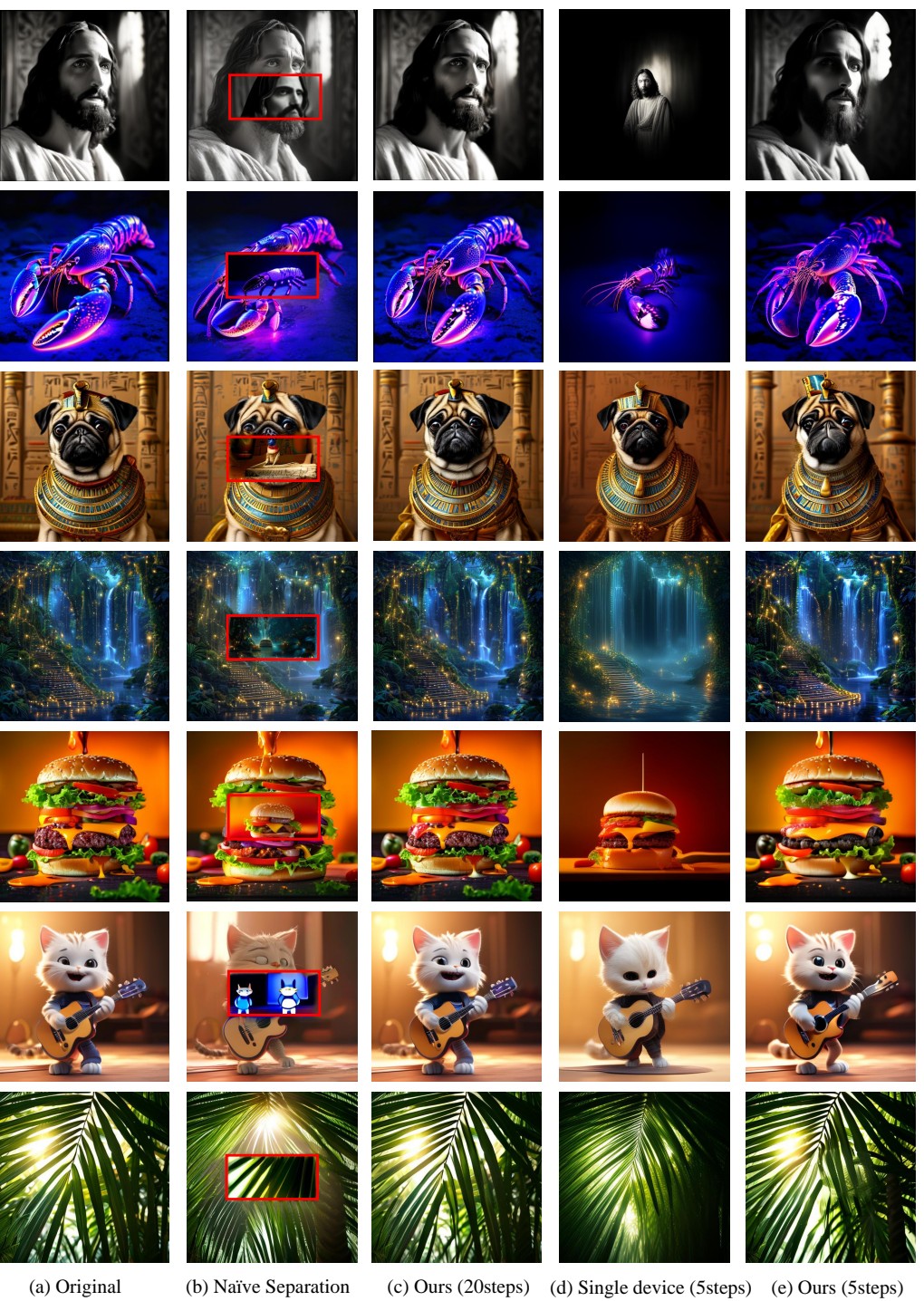

(a) Original      (b) Naïve Separation      (c) Ours (20steps)      (d) Single device (5steps)      (e) Ours (5steps)

Figure 7: More visualization results on the MJHQ-30K dataset.

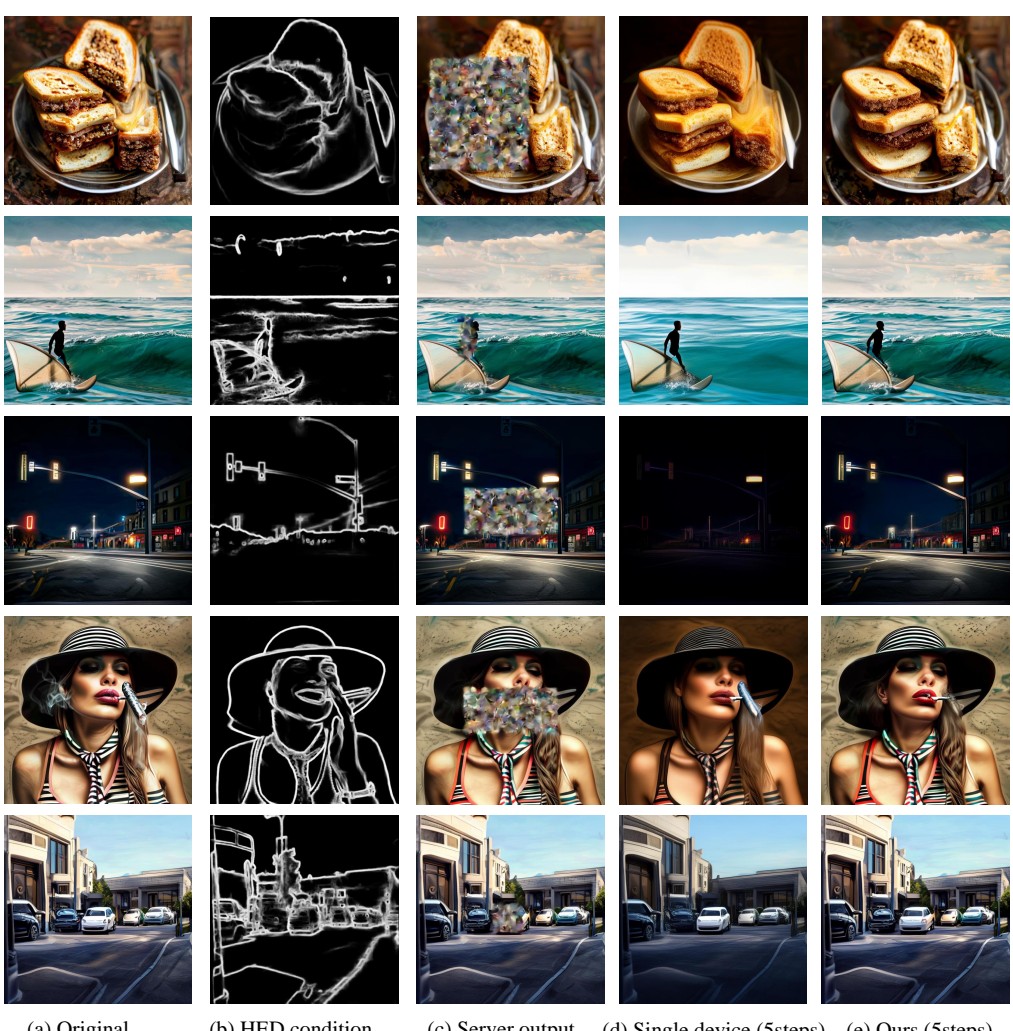

(a) Original      (b) HED condition      (c) Server output      (d) Single device (5steps)      (e) Ours (5steps)

Figure 8: More visualization results on the COCO 2017 dataset.

In contrast, the original model with 5 steps produces outputs that differ significantly from the target. This demonstrates that interaction between early and late inference steps can substantially benefit low-step models, leading to markedly improved image quality.

As depicted in Fig. 8, there are more visualization results on the COCO 2017 dataset. Our method keeps private data secure from the server and achieves far superior results to the original 5-step generation, capturing substantially more image details.

## B  IMPLEMENTATION DETAILS

The main experiments are conducted based on the SANA model. To facilitate validation, we utilize two 24GB RTX A5000 GPUs to implement and test our interactive design. Additionally, the multiply–accumulate operations (MACs) are estimated by calculating the total computational cost of DiT over multiple inference steps when generating a single image.

## C    LIMITATIONS AND BROADER IMPACT

While *PrivInfer* demonstrates strong potential in balancing privacy protection and computational efficiency through region-specific cross-device collaboration, there still remain limitations. Although the data transmission proceeds in parallel with model inference, further efficiency gains hinge upon higher transmission speeds between the client and server, which are now more attainable thanks to 5G [1] and emerging ultra-high-speed communication technologies. Moreover, applying data compression techniques offers another feasible avenue for reducing the transmission cost.

Our work advocates a new hybrid generation paradigm, which rethinks the conventional monolithic deployment of generative models. Through the separation of spatial regions and inference steps, *PrivInfer* opens up new opportunities for privacy-aware personalization, enabling real-time image editing on edge devices without compromising user privacy or requiring full model retraining. This paradigm can be extended beyond image synthesis to other tasks like video generation and personalized avatars, potentially benefiting applications in healthcare, education, and digital identity.

## D    LLM USAGE

In this work, we use ChatGPT [2] to polish our sentences and check grammar.

---

[1] https://chenweixiang.github.io/docs/R-REC-M.2083-0-201509.pdf
[2] https://chatgpt.com/

