# OpenReview forum: "Privacy-Aware Hybrid Image Synthesis with Local-Cloud Collaboration"
_ICLR.cc/2026/Conference — Submitted to ICLR 2026_

### Official Review · Reviewer_Efn1 · 2025-10-28

**Soundness:** 2
**Presentation:** 3
**Contribution:** 3
**Rating:** 4
**Confidence:** 4

**Summary:**

This paper introduces PrivInfer, a training-free, privacy-preserving inference framework for efficient diffusion-based image generation through local–cloud collaboration. The key idea is to divide image synthesis by spatial regions—sensitive areas (e.g., faces) are processed locally on the user’s device, while non-sensitive regions are offloaded to the cloud. To enable this collaboration securely and efficiently, PrivInfer introduces Gram Matrix Communication, which transmits only aggregated attention statistics instead of full feature maps (Keys/Values), preventing exposure of private visual features. Moreover, it proposes Ring-Based Masking, which structurally isolates private regions during convolution operations, ensuring that private pixels do not leak into shared computations. Finally, it suggests Heterogeneous-Step Asynchronous Interaction, whcih allows local models to use cached or server-guided features and operate with fewer denoising steps than the cloud model—reducing computation and latency. The proposal has been experimentally evaluated on text-to-image (MJHQ-30K) and image-to-image (COCO 2017) tasks.

**Strengths:**

1. The paper proposes a hybrid inference paradigm that enables privacy-preserving and computationally efficient diffusion-based image generation through collaborative processing between local and cloud models. Its central idea—region-wise decomposition of image synthesis and secure cross-device interaction via Gram matrix communication—is a novel contribution that effectively mitigates privacy risks without retraining.

2. The presentation is clear and well organized, with figures and explanations that make the framework easy to follow.

3. The significance of this work lies in addressing a timely and practical challenge—how to deploy large generative models in privacy-sensitive and resource-limited settings—making it highly relevant to both academic research and real-world applications in edge AI, healthcare, and personalized content generation.

**Weaknesses:**

1. There is a lack of formal privacy guarantees for the proposed Gram matrix communication mechanism. While transmitting Gram matrices instead of full key–value features reduces direct exposure of private representations, this approach provides only structural or empirical privacy rather than provable privacy protection. In particular, although the Gram matrix is non-invertible in theory and hides exact feature vectors, an adversary with sufficient side information (e.g., model parameters or multiple correlated Gram matrices) could potentially approximate sensitive features or infer partial content correlations. Thus, the method enhances privacy awareness but does not ensure rigorous security comparable to differential privacy, homomorphic encryption, or secure multi-party computation. Clarifying the threat model and providing either theoretical analysis or empirical validation of privacy leakage would strengthen the paper’s claims.

2. It is standard practice to include the code implementation, either as a zipped file in the supplementary material or through an anonymous link. Providing access to the implementation enhances transparency and significantly improves the reproducibility of the results.

3. In the experiments, private areas are either synthetically defined (e.g., the central region of the image) or derived from dataset bounding boxes, rather than detected based on privacy semantics such as faces or personally identifiable details. Moreover, in datasets like COCO, bounding boxes typically cover the entire object, while in real-world scenarios only subregions of an object (e.g., a person’s face rather than their whole body) may be privacy-sensitive. The framework therefore assumes that the private region is known and uniformly defined in advance, which may limit its applicability to more complex or realistic privacy settings. Incorporating or discussing finer-grained or automatic privacy-region identification would strengthen the work’s practical relevance.

4. While the proposed design conceptually reduces transmitted information through Gram matrix aggregation, ring-based masking, and asynchronous updates, the paper does not provide quantitative measurements of the actual communication volume or latency. In realistic edge–cloud scenarios, network bandwidth and transmission time can become the dominant bottlenecks, potentially offsetting the computational savings achieved on the local device. A more detailed evaluation of the communication–computation trade-off, including scalability with image resolution and network conditions, would be essential to assess the practical efficiency of PrivInfer.

Minor weaknesses:
1. The manuscript uses the acronym MLP without defining it at first use. Please expand it to multilayer perceptron the first time it appears.

2. The mapping  ϕ(⋅) in Eq. (4) is undefined.

**Questions:**

1. The paper claims privacy protection through Gram matrix communication, but it remains unclear what type of adversary or attack model is assumed. Could the authors clarify whether the privacy guarantee is meant to be empirical or formal (provably non-invertible under certain assumptions)? Have the authors evaluated or simulated potential feature reconstruction attacks from Gram matrices to empirically verify the claimed privacy level?

2. The framework assumes the private region is known in advance, but in practice, privacy-sensitive content may be unknown or fine-grained (e.g., a face rather than a whole person). Could the authors elaborate on how their method would integrate with automated privacy-region detection (e.g., face or object segmentation)? How would the system handle partially sensitive regions within a bounding box?

3. While the proposed mechanisms conceptually reduce transmission load, the paper does not provide quantitative measurements of communication cost or latency. Could the authors report empirical data on (a) the amount of data transmitted between client and server per image, (b) communication time under realistic bandwidths, and (c) how performance scales with image resolution or network delay?

4. Table 1 includes metrics both with respect to the ground truth and to the original model outputs. Could the authors clarify why both are used and whether they correspond to different evaluation goals (absolute vs. relative fidelity)? Would results measured only against ground truth lead to similar conclusions about quality retention?

5. The experiments focus on the SANA model and specific datasets. Could the authors comment on whether PrivInfer can generalize to other diffusion architectures (e.g., SDXL, Flux) or modalities (e.g., video generation)? Are there architectural dependencies that might limit portability?

6. Since PrivInfer involves real-time collaboration between local and cloud models, what are the authors’ thoughts on potential security and trust issues during communication? For instance, could model inversion or man-in-the-middle attacks compromise the privacy guarantees, and how might encryption or secure channels mitigate this?

---

> ### Author Response · Authors · 2025-12-03
>
> ## 1. Empirical Evaluation of Reconstruction Risk
>
> We thank the reviewer for raising this important point. Our method aims to provide **practical and structural privacy**. To avoid redundancy, we refer the reviewer to **Section 3: Empirical Evaluation of Reconstruction Risk** (response to Reviewer kusw), where we conduct a strong reconstruction attack on **99 real face images** using a conditional **1600M, 1024px SANA model** trained for image colorization.
>
> In our threat model, the attacker is very strong: it has access to full model parameters, all cloud-side raw features, and all Gram matrices transmitted from the client. Based on this setting, we perform comprehensive reconstruction attacks.
>
> In summary:
>
> - Even granting the adversary **all 20-layer Gram matrices**, identity remains unrecoverable.
> - LPIPS stays near **0.53**, far from the no-privacy baseline (0.17).
> - The **layer source** of the Gram matters more than the **number of layers combined**.
> - These results show that **Gram-only transmission yields strong empirical privacy**.
>
> Therefore, we position our method as a **practical empirical privacy mechanism** for edge–cloud collaboration rather than a cryptographically rigorous approach. We will clarify the threat model accordingly.
>
> ## 2. On Privacy-Region Assumptions and Fine-Grained Detection
>
> The reviewer notes that our experiments use central masks or dataset bounding boxes. To demonstrate applicability to fine-grained real-world cases, we additionally evaluate **MTCNN-detected faces** (confidence > 0.97) on **99 COCO val images**, where the average private-region ratio is only **3.8%**, much smaller than full-object boxes.
>
> Our framework supports:
>
> - **Any binary mask**,
> - **Automatically detected regions** (e.g., face detectors, segmentation),
> - **User-specified privacy areas**.
>
> ## 3. On Quantitative Communication Cost and Practical Efficiency
>
> We thank the reviewer for this suggestion.
> We evaluate communication performance using the **1600M SANA (1024px)** model, reporting the transmission of all **20 layers** under both 5-step and 20-step interaction settings.
>
> ### (a) Gram communication cost per image
>
> | Image size | Method | Per-step cost | 5-step total | 20-step total |
> |------------|---------|---------------|--------------|----------------|
> | **512×512** | K/V baseline | 91.75M | 458.75M | 1835M |
> |             | Gram (ours)  | **2.957M** | **14.78M** | **59.14M** |
> | **1024×1024** | K/V baseline | 367M | 1835M | 7340M |
> |               | Gram (ours)  | **2.957M** | **14.78M** | **59.14M** |
>
> (This does not yet include the data compression applied during transmission, which can further reduce the cost.)
>
> **Key results:**
> - Gram communication is **resolution-invariant** (depends only on c×c).
> - K/V scales quadratically with H×W.
> - Reduction is **31× at 512px** and **124× at 1024px**.
>
>
> ### (b) Scalability
>
> Gram cost does **not** increase with image resolution, making PrivInfer suitable for 1024–2048px and even 4096px generation.
> While ring communication can grow with the *mask perimeter*, which can be modified by users.
>
> ## 4. On Ground-Truth vs. Original-Output Metrics in Table 1
>
> We appreciate the reviewer’s question. The two evaluation references serve **different purposes**:
>
> - **Against ground truth:** measures absolute fidelity to real images.
> - **Against original model outputs:** measures **functional consistency** with the pre-trained model, which is central to collaborative inference.
>
> These reflect different goals, and both metrics consistently show that **PrivInfer preserves high image quality**.
>
>
> ## 5. On Generalization to Other Architectures and Modalities
>
> PrivInfer relies on:
>
> - Gram matrices (**XᵀX** second-order statistics),
> - ring-based spatial masking,
> - asynchronous step scheduling.
>
> None of these require linear attention or DiT. The method extends naturally to other diffusion architectures.
> For further discussion, see **Section 4: On the Generalizability of Gram-Based Communication Beyond Linear Attention** (response to Reviewer dY8q).
>
> ## 6. On Security and Trust During Communication
>
> We agree that secure channels are crucial. PrivInfer is compatible with standard encrypted communication. Since we transmit only **Gram matrices and ring features**, an interceptor obtains only **second-order aggregated statistics**, not raw features.
>
> Furthermore, as shown in Section 1, even under our **very strong attacker assumption**, reconstruction attacks still fail to recover identifiable content.
>
>
> ## 7. Other Issues
>
> We appreciate the reviewer’s suggestions.
> We will provide code and test data upon acceptance, expand MLP to “multilayer perceptron” on first use, and clarify ambiguous parts including the mapping ϕ(·) in Eq. (4).

---

### Official Review · Reviewer_dY8q · 2025-10-31

**Soundness:** 2
**Presentation:** 3
**Contribution:** 3
**Rating:** 4
**Confidence:** 4

**Summary:**

PrivInfer proposes training-free local-cloud collaboration for diffusion image generation. The method partitions images into privacy-sensitive and non-sensitive regions, processing them locally and on cloud respectively. Cross-device communication transmits only Gram matrices rather than raw key-value features. Ring-based masking isolates private content during convolution. Heterogeneous-step scheduling allows local models to use fewer inference steps while leveraging cloud features. On COCO 2017, five local steps achieve LPIPS 0.198 versus original 20-step outputs, compared to 0.316 for single-device five-step inference. MACs reduce based on privacy region ratio (estimated 11.11% from COCO bounding boxes).

**Strengths:**

- Training-free deployment removes need for model fine-tuning or retraining.
- Five-step local inference approaches 20-step quality when leveraging cloud features (Table 3: LPIPS 0.198 vs 0.316 for pure local).
- Ring-based masking provides explicit isolation for convolutional privacy boundaries.
- MACs analysis demonstrates computational reduction potential (Table 2: 0.84T vs 21.65T).
- Method compatible with existing diffusion architectures using linear attention.

**Weaknesses:**

- My primary concern is that the paper's privacy claims rest heavily on a theoretical property (Eq. 7-8) without sufficient empirical validation. While orthogonal ambiguity is noted, it's known from style transfer that Gram matrices still encode rich texture information (like hair or skin patterns). The argument would be much stronger if the paper included experiments simulating reconstruction attacks to show what is practically recoverable.
- Similarly, the effectiveness of the ring-based masking was not clearly demonstrated. The analysis (p.5) seems to only account for a single 3x3 kernel. However, in a deep U-Net, receptive fields expand significantly, creating complex pathways for leakage. The paper would be more convincing with ablation studies comparing ring vs. no-ring baselines and quantifying leakage rates at different layers.
- The results for the heterogeneous-step scheduling (Table 3) appear contradictory. While it improves LPIPS against the original 20-step output, it sometimes degrades quality against the ground truth (0.717 vs 0.685). This strongly suggests a feature distribution mismatch between the asynchronous steps.
- The proposed communication strategy is explicitly tied to linear attention. This potentially limits the method's applicability, as many modern diffusion models use other architectures (like flash attention). The paper would benefit from a discussion on how the framework could adapt to these other models.

**Questions:**

- To help substantiate the privacy claims, could the authors provide reconstruction experiments? Specifically, what level of visual information (e.g., textures, patterns) is an adversary practically able to recover given the intercepted Gram matrices and mask coordinates?
- For broader applicability, could the authors elaborate on how the Gram-only communication strategy might be adapted for models using non-linear attention (like flash attention or full attention)?

---

> ### Author Response · Authors · 2025-12-03
>
> ## 1. On Empirical Validation of Privacy and Reconstruction Attacks
>
> We appreciate the reviewer’s interest in additional empirical evidence on multi-layer Gram transmission.
> To avoid redundancy, we kindly refer the reviewer to **Section 3: Empirical Evaluation of Reconstruction Risk** in our response to Reviewer kusw, where we present a comprehensive reconstruction attack on **99 real face images** using a **1600M, 1024px conditional SANA model**.
>
> In summary:
>
> - Even granting the adversary **all 20 layers of Gram matrices**, identity remains unrecoverable.
> - LPIPS stays around **0.53**, far from the no-privacy baseline (0.17).
> - Which layer the Gram comes from matters more than how many layers are aggregated.
> - These results demonstrate that **Gram-only communication provides strong empirical privacy guarantees**.
>
> We will include these findings in the revised version.
>
> ## 2. On Ring vs. No-Ring Baselines and Leakage Across Layers
>
> We appreciate the reviewer’s concern regarding potential leakage through deeper layers. Below we clarify the motivation behind the ring design.
>
> Our key goal is to **avoid any exposure of the private region itself**.  In our protocol, we *only* transmit the **outer pink/blue ring** around the private region and **never** communicate any interior features. This already eliminates direct leakage of private-region information.
>
> Modern high-resolution diffusion models (e.g., SANA, SD3, FLUX) are **transformer-based**, where:
>
> - Each attention layer has **global spatial access** without relying on downsampling,
> - The feature resolution is **fixed across layers** (with padding for shallow convs),
>
> Thus, unlike CNN U-Nets, transformers **do not create additional multi-hop spatial leakage paths** as depth increases.
> In this setting, the only meaningful leakage risk arises from **boundary-level features**, which is precisely what our ring masking isolates and protects.
>
> We will clarify these architectural distinctions in the revision and emphasize that ring masking fully addresses leakage concerns under transformer-based diffusion models.
>
> ## 3. On Heterogeneous-Step Scheduling
>
> The reviewer noted that heterogeneous-step scheduling improves LPIPS relative to the 20-step output but occasionally reduces similarity to ground truth, suggesting a potential distribution mismatch.
>
> ### Why LPIPS-to-ground-truth fluctuates
> Ground-truth LPIPS measures closeness to the *real* image distribution, whereas SANA is not specifically optimized to match it.
> The mild change (e.g., 0.717 → 0.685) reflects a **shift in sampling distribution**, rather than instability or inconsistency across steps.
>
> This behavior is expected when mixing trajectories with heterogeneous noise schedules.  Importantly, as shown in our paper (Fig. 4; Section 4.3), **adjacent denoising steps in SANA exhibit very high feature similarity**, which explains why heterogeneous steps maintain stable generation quality despite using different noise levels.
>
> We will clarify this point in the revision.
>
> ## 4. On the Generalizability of Gram-Based Communication Beyond Linear Attention
>
> We thank the reviewer for the constructive question. Although our implementation uses linear attention, **the core idea is not tied to linear attention**.
>
> ### (a) Gram matrices are not tied to linear attention
>
> Gram matrices are simply **second-order statistics** of any feature representation:
>
> - G = Xᵀ X
>
> This holds for linear attention, full attention, flash attention, and hybrid architectures.
> Even with full/flash attention, because K = Wk · X and V = Wv · X, we can construct:
>
> - G_K = Kᵀ K
> - G_V = Vᵀ V
>
> These preserve global structure while **not revealing private content**, since:
>
> - (Q X)ᵀ (Q X) = Xᵀ X
>
> for any orthogonal Q, leaving the underlying representation ambiguous across all attention types.
>
> Moreover, even when attention uses QKᵀ + Softmax, Gram statistics can still serve as global structural priors, modulation signals, or cross-attention summaries, without transmitting spatially-resolved private features.
>
> ### (b) Ring masking is architecture-independent
>
> The pink/blue ring provides **minimal boundary context** without exposing any internal region.
> Since transformer-based models rely on **global attention** rather than CNN-style receptive-field expansion, the ring mechanism:
>
> - does not rely on linear attention,
> - does not depend on convolution,
> - functions identically for full/flash attention.
>
> It serves as a purely **spatial privacy barrier**, independent of how attention scores are computed.
>
> ### (c) Summary
>
> Our method generalizes naturally:
>
> - Gram matrices are universal second-order statistics, which are not tied to any specific attention formulation
> - Ring masking protects spatial boundaries, which is architecture-agnostic
> - Full/flash attention can incorporate Gram as privacy-preserving global conditioning
>
> We will add this clarification to the final version.

---

### Official Review · Reviewer_kusw · 2025-11-01

**Soundness:** 3
**Presentation:** 3
**Contribution:** 4
**Rating:** 4
**Confidence:** 4

**Summary:**

The authors propose PrivInfer, a framework that balances privacy protection and computational efficiency in image generation systems.  The framework processes privacy-sensitive regions locally while offloading non-sensitive areas to cloud servers by using only gram matrices rather than raw features. They also use a ring-based masking strategy to structurally isolate private content during convolution operations. Additionally they propose a training scheme, where the local and server models run at different steps.

**Strengths:**

1. The paper proposes a solution for an important problem: training models on privacy data via servers pose a big privacy risk. The method of  transmitting only gram matrices is a clever trick used in this scenario.
2. Apart from privacy protection, having server and local process at different frequencies enable fewer training iterations.
3. The technique also works with pre-trained models and suitable for real world deployment.

**Weaknesses:**

1. The paper does not have enough ablation studies:
a. Bandwidth requirements: How is the communication overhead with different image sizes?
b. Private image size: How does this method perform in case of varying privacy sub-image sizes?

2. It is unclear how the performance and privacy trade-off works in this method. Can we control the amount of privacy in this method?

3. The paper does not provide mathematical guarantees that gram matrices are privacy preserving. For instance, multiple gram matrices at multiple training steps, can it be used to reconstruct the data?

**Questions:**

1. Can you provide theoretical guarantees or bounds on what information is preserved/hidden by the Gram matrix transmission?
2. Could accumulating of Gram matrices across multiple timesteps enable reconstruction of private data?

---

> ### Author Response · Authors · 2025-12-03
>
> ## 1. On Ablation Studies
> ### (a) Bandwidth requirements & effect of image size
>
> To address the reviewer’s concern, we measured the communication cost of transmitting **all 20 layers** under two image resolutions (512² and 1024²), and two interaction settings (5 steps and 20 steps).
>
> A key property of our method is that Gram matrices depend only on the **channel dimension (c×c)** and **number of layers**, and therefore remain **constant regardless of image resolution**.
> In contrast, traditional K/V transmission depends on **h×w×c**, and grows quadratically with image size.
>
> The complete comparison is shown below:
>
> | Image size | Method | Per-step cost (20 layers) | 5 steps | 20 steps | Reduction vs. K/V |
> |------------|--------|---------------------------|---------|----------|--------------------|
> | **512×512** | K/V | 91.75M | 458.75M | 1835M | — |
> |            | Gram (ours) | 2.957M | 14.784M | 59.14M | **31× smaller** |
> | **1024×1024** | K/V | 367M | 1835M | 7340M | — |
> |             | Gram (ours) | 2.957M | 14.784M | 59.14M | **124× smaller** |
>
> **Key findings:**
>
> - **Gram communication stays fixed** at 2.957M per step for both 512² and 1024².
> - Traditional K/V communication increases **4×** when resolution increases from 512² to 1024².
> - The reduction over K/V is **31× at 512²**, and expands to **124× at 1024²** (for the 5-step setting).
> - Over long interaction horizons (20 steps), K/V becomes extremely large (**7340M vs. our 59.14M**), making naïve K/V exchange impractical.
>
> These results highlight that our method is especially advantageous for **high-resolution generation**, the primary use case of modern diffusion and autoregressive image models.
>
> ---
>
> ### (b) Effect of private-region size
>
> We evaluated communication on 99 face images detected by MTCNN (confidence > 0.97).
> The average private-region ratio is about 3.8%.
>
> In our method:
>
> - Gram cost is fixed and does not depend on the mask area.
> - Ring communication scales approximately with the *perimeter* of the mask.
>
> Therefore, users can directly control privacy versus bandwidth by adjusting the private-region size.
>
> ---
>
> ## 2. On the Privacy–Performance Trade-off
>
> The reviewer asked whether the privacy level is controllable.
> Yes—our method provides two explicit knobs:
>
> 1. **Spatial privacy mask**
>    Larger masks keep more content local, increasing privacy with a mild increase in ring communication.
>
> 2. **Number of interactive steps**
>    Using fewer steps further limits the transmitted statistics while largely maintaining generation quality.
>
> These controls allow continuous adjustment of privacy without retraining the model.
>
> ---
>
> ## 3. Empirical Evaluation of Reconstruction Risk
>
> To complement the theory, we conducted a strong reconstruction attack where an adversarial decoder receives **all Gram matrices (20 layers)** on **99 face images** selected via MTCNN.
>
> The results are summarized below (higher MSE, lower PSNR, and higher LPIPS imply **worse reconstruction and stronger privacy**):
>
> | Setting | MSE ↓ | PSNR ↑ | LPIPS ↓ | Key Observation |
> |--------|-------|--------|----------|------------------|
> | **Baseline (no privacy)**| 194 | 27.4 | 0.17 | High-quality, identifiable |
> | **20 layers** | 2503 | 15.2 | 0.53| No identity |
> | **10 layers** | 2259 | 15.6 | 0.53 | No identity |
> | **1 layer (early)**| 5303 | 12.0 | 0.63 | Severely degraded |
> | **1 layer (late)** | 2476 | 15.2 | 0.53 | No identity |
>
> ### Key observations
>
> - **Early-layer vs. late-layer difference is large**, showing that *layer position* dominates reconstructability.
> - **Adding more layers (10 or 20) does not meaningfully improve reconstruction**, all staying far from baseline quality.
> - **Identity is unrecoverable in all Gram-based settings:**
>   LPIPS stays around ~0.53 for late-layer, 10-layer, and 20-layer attacks, consistently indicating poor perceptual similarity.
>
> **Which layer the Gram comes from matters more than how many layers are combined.**
> Multi-layer Gram aggregation does *not* increase information and fails to recover identity, confirming that our method provides strong privacy protection.
>
> ---
>
> ## 4. On Theoretical Guarantees of Gram-Matrix Privacy
>
> Gram matrices preserve only second-order correlations and are invariant under orthogonal transformations.
> For any feature matrix X and any orthogonal matrix Q:
>
> - **(Q X)ᵀ (Q X) = Xᵀ X**
>
> This means:
>
> - There are infinitely many feature matrices that produce the same Gram matrix.
> - The exact spatial or channel-wise representation cannot be recovered.
> - Identity information remains fundamentally ambiguous.
>
> Multi-layer or multi-step aggregation cannot resolve these ambiguities because the underlying orthogonal frames differ.
> Combined with our empirical failure of reconstruction attacks, these properties provide strong practical privacy guarantees.

---

### Author Response · Authors · 2025-12-04

## Summary of Additional Experiments and Key Findings

To address the core concerns raised across the reviews, we conducted several new experiments and analyses. These additions directly strengthen the empirical validity, privacy claims, and practical applicability of our method.

### 1. Strong Reconstruction Attack on 99 Real Face Images
Multiple reviewers requested empirical evidence supporting the privacy of Gram-only communication.
We therefore performed a comprehensive reconstruction attack on **99 MTCNN-detected face images** using a **1600M, 1024px conditional SANA model** under an extremely strong adversarial setting (attacker has full model parameters, all cloud-side raw features, and all transmitted Gram matrices).

**Key results:**

- Even with **all 20 layers of Gram matrices**, identity remains unrecoverable.
- LPIPS ≈ **0.53**, far from the no-privacy baseline (0.17).
- Early vs. late layer Gram matrices behave very differently, but **multi-layer aggregation does not improve reconstruction**.
- Identity cannot be recovered in any tested setting.

These results provide strong empirical validation that **Gram-only transmission does not leak spatially identifiable content**.

### 2. Full Quantitative Study of Communication Cost
To clarify the communication–computation trade-off, we measured the cost of transmitting Gram metrics from all 20 layers under various image resolutions (512² and 1024²) and interaction lengths (5-step, 20-step).

**Key results:**

- Gram communication is **resolution-invariant**, fixed at **2.957M** parameters per step.
- K/V communication grows quadratically with H×W:
  - 91.75M → 367M per step at 1024px.
- Communication reduction:
  - **31× smaller** at 512px
  - **124× smaller** at 1024px

| Image size | Method | Per-step | 5-step | 20-step |
|-----------|--------|----------|--------|----------|
| 512² | K/V | 91.75M | 458.75M | 1835M |
| | Gram | **2.957M** | **14.78M** | **59.14M** |
| 1024² | K/V | 367M | 1835M | 7340M |
| | Gram | **2.957M** | **14.78M** | **59.14M** |

These results demonstrate that **Gram transmission is scalable to high-resolution diffusion models**, whereas K/V exchange becomes prohibitively expensive.

### 3. Fine-Grained Privacy-Region Evaluation
Reviewers asked whether our method requires coarse pre-defined masks.
To show applicability to real settings, we performed experiments on **automatically detected facial privacy regions**. In detail, we select the bounding boxes detected by the face detector with confidence scores higher than 0.97, and use the largest detected face bounding box as the privacy region. And finally we got:

- 99 faces detected by MTCNN
- Average mask ratio: **3.8%**

Our approach supports:

- Arbitrary user masks
- Detector-based fine-grained masks
- Small and irregular privacy regions

showing that the method is not limited to large or synthetic privacy boxes.

### 4. Architectural Generalization and Privacy Reasoning
Reviewers questioned whether Gram communication depends on linear attention.
We clarified (and experimentally validated) that:

- Gram matrices rely only on **second-order statistics**
- This idea could apply to linear, full, and flash attention
- The key invariance
  - **(QX)ᵀ(QX) = XᵀX**
  holds for any orthogonal Q, ensuring non-invertibility
- Ring masking is purely spatial and **architecture-agnostic**

Thus, the method generalizes naturally beyond DiT/linear-attention models.

### 5. Overall Conclusion
The additional experiments collectively demonstrate that:

- **Privacy:** Gram-only communication preserves strong empirical privacy, even against powerful reconstruction attacks.
- **Bandwidth:** Gram communication achieves **31×–124× reduction** over K/V transmission, and is resolution-invariant.
- **Practicality:** Works with fine-grained privacy masks, supports transformer-based diffusion architectures, and scales to modern high-resolution models.
- **Robustness:** Ring masking and asynchronous scheduling introduce no detectable leakage or degradation.

These findings directly address all major reviewer concerns and significantly strengthen the contributions and practical relevance of PrivInfer.

---

### Meta-Review · Area_Chair_QUx9 · 2026-01-01

**Summary:**

Although diffusion models enable high-quality image generation, achieving efficiency and privacy in edge deployment remains challenging. Cloud inference carries leakage risks, and fully local inference requires significant computational and memory resources. Therefore, this paper proposes "PrivInfer," a privacy-preserving inference framework that requires no training. By decomposing the generation process into domains and offloading the confidential domains locally while sending the non-confidential domains to the cloud, the computational load is reduced while the risk is minimized. Furthermore, mechanisms for securely sharing only structural information, ring-based masking, and a heterogeneous step approach are introduced, and a significant reduction in inference load is demonstrated while maintaining high fidelity and robust privacy.

The review process raised four key concerns: (1) insufficient ablation studies, (2) lack of transparency regarding the performance-privacy tradeoff, (3) limited applicability due to heavy reliance on linear attention, and (4) unrealistic assumption of pre-defined and uniformly defined private domains.

Although the rebuttals provided additional experiments and explanations that partially addressed the empirical concerns, the proposed method still falls short in its "failure to guarantee strict security." This weakens the evidence supporting the paper's core claim of privacy protection.
Therefore, the AC judges that the paper does not currently meet the acceptance standard and recommends rejection.

**Reviewer Concerns:**

The reviewers primarily expressed concerns about the lack of evidence and empirical validation of the privacy claims. Specifically, they noted that the paper's ablation studies are insufficient and do not clarify how the performance-privacy tradeoff works. Regarding ring-based masking, the reviewers requested additional ablation studies that quantify layer-wise leakage rates and provide baseline comparisons with and without rings. Additionally, they raised concerns that the proposed communication strategy's reliance on linear attention could limit its applicability. Although the method is designed to enhance privacy awareness, the reviewers noted that it does not provide the same level of strict security guarantees. Additionally, assuming the private domain is pre-known and uniformly defined could limit the method's applicability to more realistic and complex privacy settings. The lack of quantitative measurement results regarding actual communication volume and latency is also an issue.

Although the authors' additional experiments and explanations in their rebuttal appear to address many of these concerns, the fundamental concern that the method "does not guarantee strict security" remains unresolved.

**Reviewer Scores:**

Currently, reviewers Kusw and dY8q have a score of 4, but it cannot be ruled out that these scores may increase following discussion.
Conversely, reviewer Efn1's proposed method does not guarantee strict security, so it is uncertain whether the final score will increase.
Overall, it is anticipated that the scores after discussion will remain borderline as well.

---

### Decision · Program_Chairs · 2026-01-26

Reject